# Glycolate combats massive oxidative stress by restoring redox potential in *Caenorhabditis elegans*

Veronica Diez[1], Sofia Traikov[1], Kathrin Schmeisser[1], Akshay Kumar Das Adhikari[1] & Teymuras Vakhtang Kurzchalia [1✉]

Upon exposure to excessive reactive oxygen species (ROS), organismal survival depends on the strength of the endogenous antioxidant defense barriers that prevent mitochondrial and cellular deterioration. Previously, we showed that glycolic acid can restore the mitochondrial membrane potential of *C. elegans* treated with paraquat, an oxidant that produces superoxide and other ROS species, including hydrogen peroxide. Here, we demonstrate that glycolate fully suppresses the deleterious effects of peroxide on mitochondrial activity and growth in worms. This endogenous compound acts by entering serine/glycine metabolism. In this way, conversion of glycolate into glycine and serine ameliorates the drastically decreased NADPH/NADP$^+$ and GSH/GSSG ratios induced by $H_2O_2$ treatment. Our results reveal the central role of serine/glycine metabolism as a major provider of reducing equivalents to maintain cellular antioxidant systems and the fundamental function of glycolate as a natural antioxidant that improves cell fitness and survival.

[1] Max Planck Institute of Molecular Cell Biology and Genetics, Dresden, Germany. ✉email: kurzchalia@mpi-cbg.de

Oxidative stress is a metabolic state defined as a disturbance in the balance between the production of reactive oxygen species (ROS) and antioxidant defenses in a living organism[1,2]. ROS are highly reactive molecules due to their unpaired electrons and are produced as a result of normal cellular metabolism. However, when produced at high concentrations, they can lead to destruction of cellular structures, lipids, proteins, and genetic material[1,3,4], contributing to many pathological conditions, including cancer, neurological disorders, hypertension, and diabetes[5–10].

The electron transport system of mitochondria is one of the major producers of ROS and thus is particularly susceptible to attack by them. In fact, overproduction of ROS may lead to a decrease in respiratory enzyme activities and mitochondrial membrane potential (MMP)[11–13]. One of the most commonly used toxins to model response and sensitivity to oxidative damage in mitochondria is the bipyridyl herbicide paraquat ($PQ^{2+}$)[14,15]. $PQ^{2+}$ is a redox cycling compound that takes electrons from the electron transport chain or NADPH and, via its cationic radical, reduces molecular oxygen to the superoxide radical $O_2^{\cdot-}$. The latter has pleiotropic effects on mitochondria. Firstly, it can cause direct damage to a plethora of cellular iron-containing targets (including the enzyme aconitase and mitochondrial electron-transport complexes carrying Fe–S clusters). Secondly, it is converted into $H_2O_2$, and together with excessive iron, this produces the extremely aggressive hydroxyl radical, which is able to interact with any chemical bond in its vicinity[4,11,16]. To protect biological systems from free radicals, several cellular antioxidant barriers exist, including enzymatic and non-enzymatic ones. Among the major defense mechanisms against ROS, and in particular against $H_2O_2$, is the tripeptide glutathione. This most abundant cellular ROS scavenger exists in reduced (GSH) and oxidized states (GSSG)[11,17,18].

In a previous report, we showed that the paraquat-mediated decline in MMP can be overcome by supplementation with naturally produced glycolic acid (GA) in both HeLa cells and *Caenorhabditis elegans*[19]. This compound also increased in vitro survival of primary dopaminergic neurons from a mouse model of Parkinson's disease. The mechanism of action of GA, however, remained elusive. Here, we report that GA supplementation exerts a protective role in *C. elegans* worms exposed to oxidative stress caused by $H_2O_2$ treatment. These beneficial effects require the incorporation of GA into glycine/serine metabolism. We demonstrate that the restoration of mitochondrial functions is abolished in animals unable to metabolize GA. Furthermore, we show that GA supplementation increases the NADPH/NADP⁺ ratio, in this way stimulating the regeneration of reduced glutathione, resulting in a sharp improvement in worm development and lifespan. Thus, GA restores the redox potential of the cell acting as a natural antioxidant.

## Results

### Glycolate restores paraquat-mediated reduction in MMP via a GSH-dependent mechanism

It has been widely established that major aspects of $PQ^{2+}$ toxicity towards mitochondria are the reduction in oxygen consumption rate (OCR) and MMP[14,15]. Indeed, when *C. elegans* larvae are exposed to 200 μM $PQ^{2+}$ for 2 days, membrane potential drops almost two-fold (Fig. 1a). As previously reported, supplementation with 10 mM GA in $PQ^{2+}$-treated animals, restores the MMP (Fig. 1a)[19].

To understand the molecular mechanisms of GA action, we first asked whether its protective effect depends on the functionality of the *C. elegans* antioxidant machinery. For this, we decreased the levels of glutathione, which is a key antioxidant molecule in the cell. This was achieved using buthionine sulfoximide (BSO), a specific inhibitor of GSH synthesis[20,21] (Fig. 1b). In the presence of BSO, GA had no positive effect on the MMP (Fig. 1b). Addition of GSH to BSO-treated worms, however, restored the MMP (Fig. 1b). Uptake of GSH by worms was demonstrated by measuring it under different conditions (Supplementary Fig. 1a). Levels of the antioxidant were decreased upon BSO treatment but were restored when GSH was included in the medium observing, at the same time, a correlation between the tripeptide amounts and MMP (Supplementary Fig. 1b). With this, our results indicate that the effect of GA depends on the activity of endogenous antioxidant systems and in particular on GSH.

Reduced mitochondrial activity induced by $PQ^{2+}$ exposure leads to slowing down of growth. However, in contrast to its beneficial effect on the MMP, GA was not able to rescue neither the OCR nor the growth delay in response to $PQ^{2+}$ (Supplementary Fig. 2a, b). This is consistent with the pleiotropic effects of this herbicide. Firstly, $PQ^{2+}$ is consuming NADPH by the redox cycling and thus can diminish several anabolic pathways requiring this cofactor (i.e. biosynthesis of fatty acids, lipoprotein and amino acids). Secondly, toxicity of $PQ^{2+}$ can be explained by production of superoxide that has numerous targets in the cell, among them iron-containing enzymes.[11,14,22–25]

### Glycolate counteracts the deleterious effects of $H_2O_2$ on mitochondrial function, animal reproductive capacity, and lifespan

Superoxide, can directly inactivate some Fe–S cluster-containing enzymes (e.g. aconitase) but it can also be transformed into another ROS molecule, $H_2O_2$. The latter could be responsible for some of the effects observed after $PQ^{2+}$ exposure. We therefore sought to understand which of these negative effects is ameliorated by GA treatment. Firstly, we quantified $H_2O_2$ levels upon exposure of *C. elegans* larvae to $PQ^{2+}$. In agreement with previous reports[26–28], treatment of *C. elegans* larvae with the herbicide induced a dramatic elevation of $H_2O_2$ (Fig. 1c). In contrast, in animals co-incubated with GA, peroxide levels were similar to those of untreated worms (Fig. 1c). However, when the inhibitor of GSH synthesis, BSO, was included in the medium, levels of $H_2O_2$ were comparable to those obtained in the absence of GA (Fig. 1c). Furthermore, addition of GSH to BSO-treated worms significantly reduced the amounts of peroxide (Fig. 1c). These results indicate that GA decreases $H_2O_2$ levels and this effect requires a functional antioxidative machinery.

We assumed then that the $H_2O_2$ produced by $PQ^{2+}$ is the main contributor to the observed decrease in MMP. Indeed, when worms were treated with peroxide, there was a drop in MMP, but coincubation with GA prevented this effect (Fig. 2a). Similar to $PQ^{2+}$, $H_2O_2$ also induced a decline in the respiration rate and a concomitant growth delay (Fig. 2b, c). Remarkably, unlike with $PQ^{2+}$ treatment, in this case GA was able to restore both traits (Fig. 2b, c). Again, BSO treatment prevented the GA-mediated improvement in MMP, OCR, and growth rate, thus strongly indicating that glutathione synthesis is fundamental for the observed oxidative stress relief mediated by GA (Fig. 2a–c). Unexpectedly, GSH supplementation in these conditions was deleterious for the worms and therefore, it was not possible to analyze the effect of this metabolite.

Notably, GA supplementation also ameliorated the shortened lifespan and reduced reproductive capacity caused by $H_2O_2$. Animals exposed to $H_2O_2$ during larval stages and transferred to control plates as young adults showed a significantly reduced lifespan (Control worms: 18.75 ± 1.13 days; $H_2O_2$ worms: 5.88 ± 1.47 days [$p = 0.0004$]). Addition of GA significantly increased lifespan (14.75 ± 0.59 days [$p = 0.0014$]) making it comparable to that of untreated worms (Fig. 2d and Supplementary Data 1). A

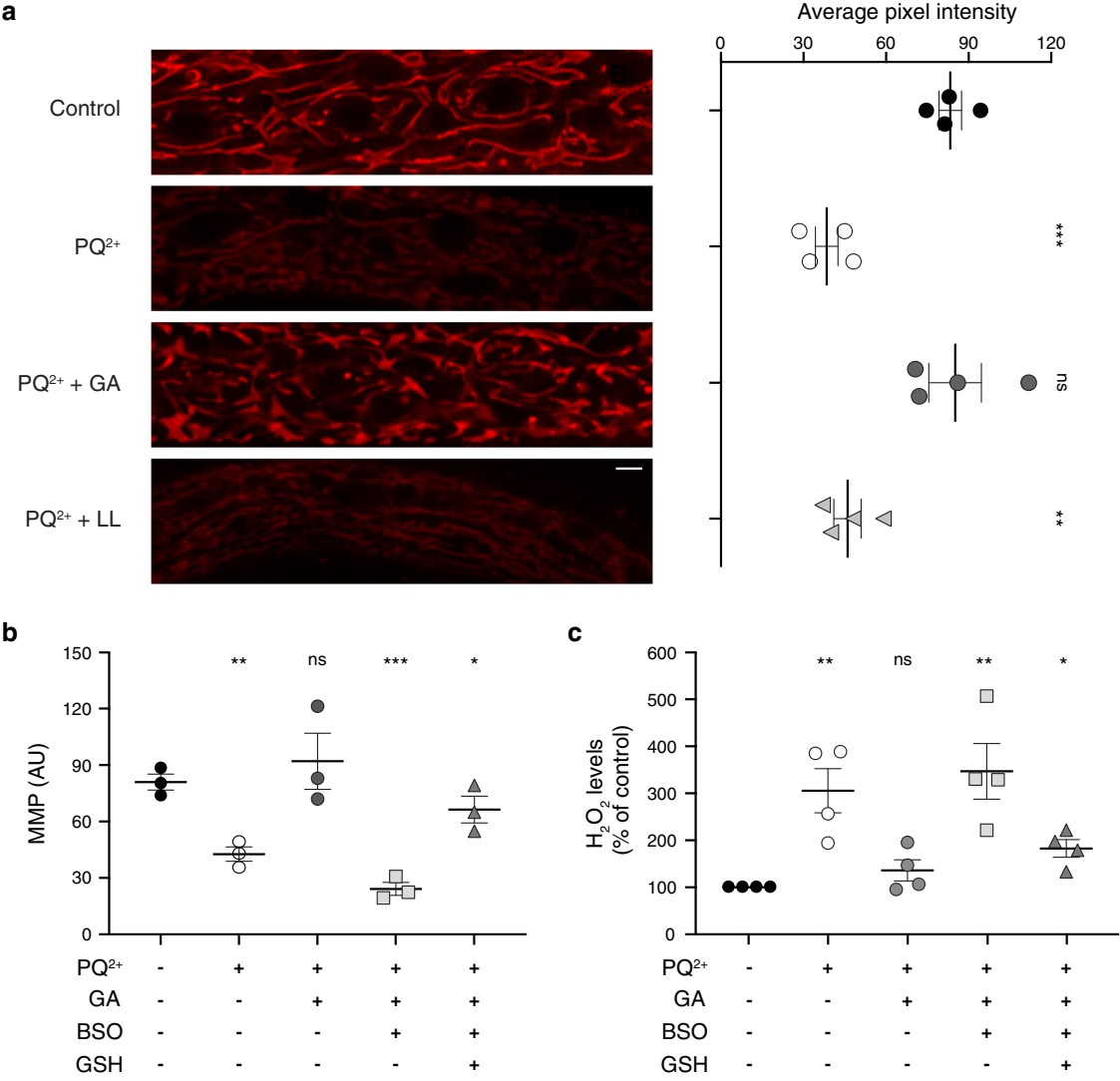

**Fig. 1 Glycolate supplementation rescues mitochondrial membrane potential and counteracts oxidative stress of paraquat-treated worms through a glutathione-dependent mechanism. a** Mitochondrial membrane potential ($\Delta\Psi$m-MMP) after 2 days of $PQ^{2+}$ treatment supplemented or not with glycolate or L-lactate. Panels on the left display mitochondrial fluorescence intensity (scale bar 10 μm) with the corresponding quantification on the right ($n = 4$). **b** Effect of inhibition of glutathione synthesis or glutathione supplementation on mitochondrial membrane potential ($n = 3$). In **a** and **b** fluorescence intensities were recorded after MitoTracker Red CMXROS staining. **c** $H_2O_2$ levels in *C. elegans* worms treated in the same conditions as in (**b**). After 2 days of treatment, $H_2O_2$ accumulation was measured with AmplexRed and fluorescence intensity was normalized to protein levels ($n = 4$). Error bars represent standard error of the mean. 200 μM paraquat ($PQ^{2+}$), 10 mM glycolate (GA), 10 mM L-lactate (LL), 0.5 mM BSO, and 1 mM GSH were used as indicated.

similar positive effect was observed when analyzing brood size and embryonic lethality in animals that had been treated only with $H_2O_2$ or in combination with GA (Fig. 2e, f, respectively).

**Glycolate supplementation prevents the drop in GSH/GSSG ratio caused by $H_2O_2$ exposure.** Glutathione is a tripeptide composed of glycine, cysteine, and glutamate that coexists in two interconvertible forms: a reduced one, GSH, and an oxidized glutathione disulfide (GSSG), which is produced upon interaction with oxidative molecules. To test if GSH and GSSG concentrations are affected when treating *C. elegans* larvae with $H_2O_2$ and GA, we quantified glutathione levels upon exposure to peroxide (Fig. 3). We observed that $H_2O_2$ treatment caused a significant decrease in total glutathione compared to non-treated animals, but this parameter was partially restored in the presence of GA (51% [$p = 0.0457$] and 91% [$p = 0.1231$] of the control,

respectively). The effect of GA was again lost, however, if the inhibitor BSO was included in the medium (65% of the control [$p = 0.0051$]) (Fig. 3a). More notable was the dramatic increase in the oxidized form of glutathione caused by $H_2O_2$, an effect that was also prevented by addition of GA (513% [$p = 0.0253$] and 140% [$p = 0.0778$] of the control, respectively) (Fig. 3b).

One of the most common markers of oxidative stress and the extent of damage caused by it is the ratio of the reduced to the oxidized form of glutathione (GSH/GSSG)[29–31]. Remarkably, we obtained a 10-fold reduction in this parameter in worms incubated with peroxide alone but only a 28% decrease if GA was included in the medium (GSH/GSSH per worm = 107.5 in control; 11.3 in $H_2O_2$ [$p = 0.0078$] and 77.4 in $H_2O_2$ + GA [$p = 0.103$]). The effect of the latter was partially prevented by inhibition of glutathione synthesis with BSO (GSH/GSSG per worm = 36 [$p = 0.012$]) (Fig. 3c), in agreement with previously reported alterations of redox homeostasis caused by this inhibitor[32–34]. These

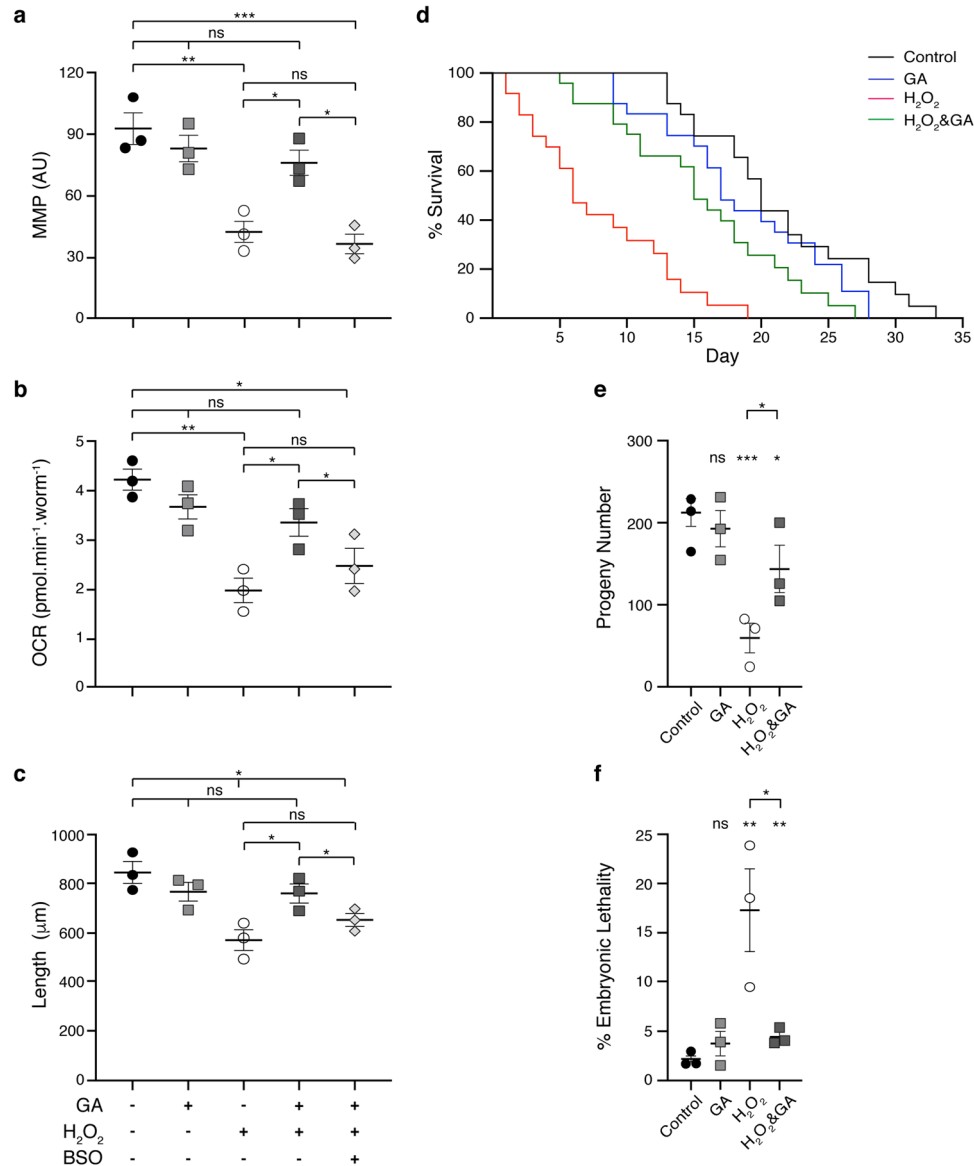

**Fig. 2 Glycolate supplementation maintains mitochondrial function and improves animal physiology of peroxide-treated *C. elegans* worms. a** Mitochondrial membrane potential (ΔΨm-MMP) was measured in the different strains after 3 days of treatment via MitoTracker Red CMXROS staining and recording the fluorescence intensities. **b** Respiration rates in terms of oxygen consumption rate (OCR) determined after 4 days of treatment using a Seahorse Analyzer and normalized to number of worms. **c** Size of worms after 4 days of treatment, where the lengths in µm of at least 20 animals were averaged per condition in each experiment. In **d–f**, life span analysis without FUdR, brood size, and embryonic lethality, respectively, is shown for worms that had been exposed to $H_2O_2$ and 10 mM GA, as indicated, during larval development. Error bars represent standard error of the mean ($n = 3$).

results clearly demonstrate that GA exerts its action by modulating a major antioxidant defense parameter, namely the ratio of GSH to GSSG. Moreover, we found that the accumulation of $H_2O_2$ was diminished in animals co-supplemented with peroxide and GA in comparison to those treated with peroxide alone, but unchanged if glutathione synthesis was blocked by BSO (Fig. 3d).

**Oxidation of glycolate is a prerequisite for its activity**. A logical question that arises is how treatment with GA can lead to an increase in GSH/GSSG ratio. We therefore explored the underlying metabolic pathways that could link alterations in glutathione levels to GA itself or some of its downstream metabolites. Metabolization of GA in animals has been reported to occur mainly via its oxidation to glyoxylate (Fig. 4)[35–37], which can subsequently be transaminated to form glycine. In turn, glycine can undergo many transformations, among them, entering one-carbon metabolism

(by donating a methyl group to tetrahydrofolate, THF) and the production of cysteine via conversion to serine (Fig. 4, KEGG pathway database).

To test whether glycolate needs to be metabolized to exert its antioxidant activity, we sought to obtain a mutant unable to catalyze the oxidation of glycolate to glyoxylate. In *C. elegans*, there are three enzymes that can be potentially responsible for this reaction (Fig. 4). The corresponding genes are: F41E6.5 (*gox-1*), encoding the ortholog of the human glycolate oxidase (HAO-1); *ldh-1* (encoding the lactate dehydrogenase), which has been found to have glyoxylate reductase activity[38,39]; and C31C9.2. The latter is predicted to encode the phosphoglycerate dehydrogenase involved in the biosynthesis of serine, but its sequence displays the highest homology to the human glyoxylate reductase/hydroxypyruvate reductase (Supplementary Fig. 3), reported to catalyze the interconversion of glyoxylate and glycolate[40–43] (Fig. 4). From

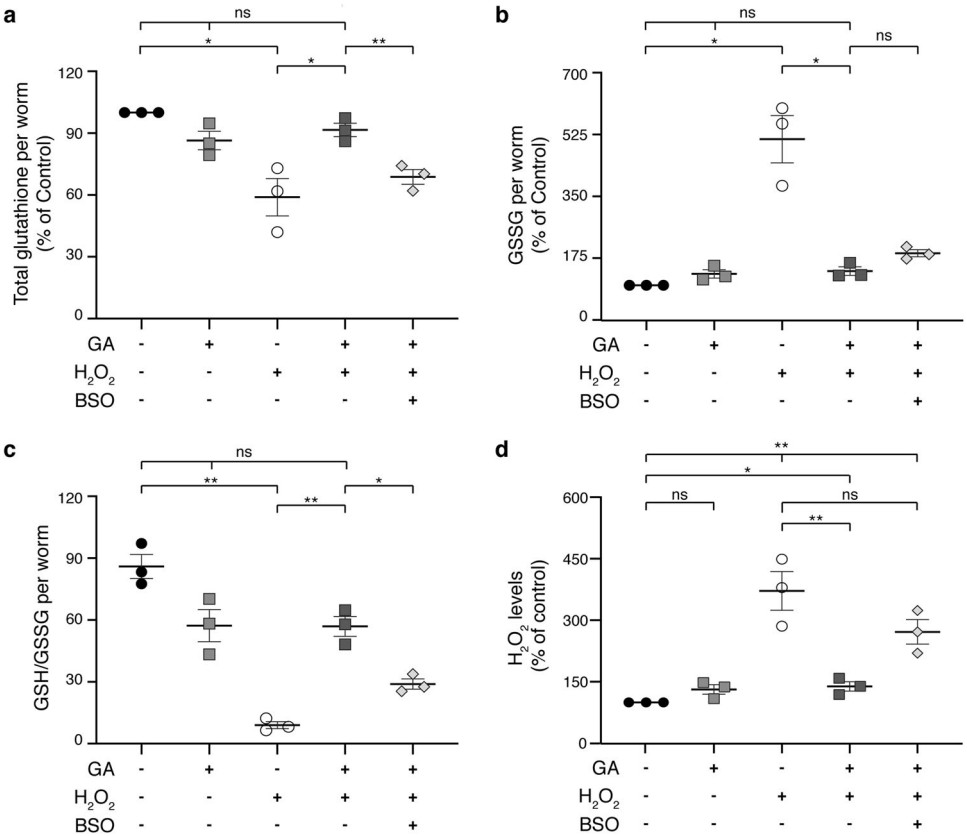

**Fig. 3 Glycolate addition prevents the drop in the antioxidant capacity of glutathione caused by peroxide. a** Total glutathione (GSH + GSSG), **b** GSSG, **c** ratio of GSH to GSSG levels, and **d** $H_2O_2$ levels in animals exposed to 100 mM $H_2O_2$ and supplemented or not with glycolate and BSO. Values were determined after 3 days of treatment and normalized to number of worms. In **d** $H_2O_2$ accumulation was measured with AmplexRed and fluorescence intensity was normalized to protein levels. 100 mM $H_2O_2$, 10 mM glycolate (GA) and 0.5 mM BSO were used as indicated. Error bars represent standard error of the mean ($n = 3$).

here, this gene will be designated as *ghpr-1* (glyoxylate/hydroxypyruvate reductase-1).

Single mutants of *gox-1* and *ldl-1*, or their double mutants showed no reduction in the effect of glycolate on MMP, OCR, or growth. The deletion mutant of *ghpr-1*, however, displayed a significantly reduced effect of glycolate on MMP, OCR and developmental rate upon $H_2O_2$ treatment (Fig. 5a–c). Remarkably, the glycolate-mediated protection was totally abolished in the *ghpr-1; gox-1; ldh-1* triple mutant strain, demonstrating the overlapping activities of these three enzymes (Fig. 5a–c).

To test whether the reduction or absence of positive effects of GA on these mutants is due to deficient ROS-scavenging abilities, relative glutathione levels were quantified. Interestingly, both *ghpr-1* and *ghpr-1; gox-1; ldh-1* strains exhibited decreased levels of total glutathione compared to untreated N2 worms, a situation that was worsened by peroxide treatment but, in contrast to wild-type animals, unaffected by the addition of GA (Fig. 6a). Additionally, the significant increase in the oxidized form of the antioxidant, GSSG, observed in animals treated with peroxide, was not rescued by co-incubating the mutants with GA (Fig. 6b). A similar lack of response to GA was observed when analyzing the ratio of GSH to GSSG (Fig. 6c). These results clearly demonstrate that oxidation of GA is catalyzed mostly by GHPR-1 but also by GOX-1 and LDL-1, and suggest that further metabolization of this compound is crucial for its protective role on mitochondrial function and growth upon oxidative damage.

**Relief of oxidative stress requires entry of glycolate into serine–glycine metabolism.** According to the scheme in Fig. 4,

glycolate can be converted into glycine and L-serine via the serine–glycine metabolic pathway. We asked whether these amino acids could also have a positive influence on *C. elegans* larvae exposed to peroxide. Indeed, both glycine and L-serine were able to restore MMP, respiration, growth, and GSH/GSSG ratio to control levels when added to $H_2O_2$-treated worms (Figs. 5 and 6). Moreover, both glycine and L-serine supplementation could overcome the deleterious effects of peroxide on strains deficient in the oxidation of glycolate (single *ghpr-1* and *ghpr-1; gox-1; ldh-1* triple mutants). Thus, these compounds are downstream effectors in the metabolic pathway of GA, as depicted in Fig. 4.

The next step was to understand the molecular mechanisms underlying the action of these two amino acids. Glycine is a component of the tripeptide glutathione, but it can also be converted into L-serine, which is a precursor of L-cysteine, another building block of GSH. However, more important perhaps is the conversion of glycine to L-serine that feeds one-carbon metabolism with THF as a key player (Fig. 4). Transformation of glycine into L-serine requires two enzymatic activities: one catalyzed by serine hydroxymethyltransferase (SHMT, encoded by *mel-32*) and the other by the glycine cleavage system (GCS, composed of the subunits T, P, L, and H) (Fig. 4)[44–46]. Through the action of GCS, glycine can provide a methyl group for the production of 5,10-methylene-$H_4$ folate (5,10-$CH_2$-THF), which together with another unit of glycine is a substrate of SHMT for the synthesis of L-serine (Fig. 4).

We tested the action of GA, glycine, and L-serine on worms upon knock down of the SHMT and the GCS activities by RNAi

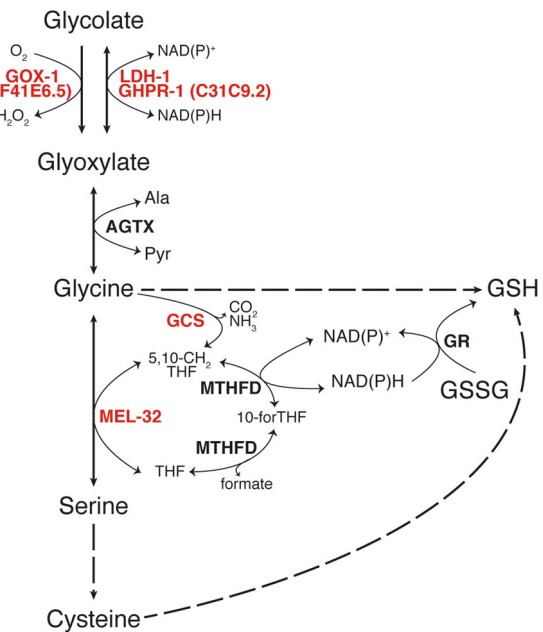

**Fig. 4 Metabolic pathway of glycolate in *C. elegans*.** In red, genes used in this work. In bold, enzymes catalyzing each reaction. Dashed arrows indicate more than one enzymatic step. This diagram was constructed according to "glyoxylate and dicarboxylate metabolism" (cel00630), "glycine, serine and threonine metabolism" (cel00260) and "one-carbon pool by folate" (cel00670) from the KEGG pathway database and the metabolic network of glycolate, glycine, and serine in WormFlux[46,81]. GOX-1 glycolate oxidase 1, LDH-1 lactate dehydrogenase 1, GHPR-1 glyoxylate reductase/hydroxypyruvate reductase 1, AGTX alanine-glyoxylate transaminase, Ala L-alanine, Pyr pyruvate, GCS glycine cleavage system, MEL-32 serine hydroxymethyl transferase, MTHFD methylene tetrahydrofolate reductase, GR glutathione reductase, 5,10-CH$_2$THF 5,10-methylen tetrahydrofolate, 10-forTHF 10-formyl tetrahydrofolate, THF tetrahydrofolate.

against *mel-32* or *gcst-1* (encoding SHMT and subunit T of the GCS in *C. elegans*, respectively) (Fig. 4). Remarkably, none of these three compounds were able to restore either MMP, OCR, or growth rate, in contrast to the restored parameters exhibited in worms fed with empty vector RNAi bacteria (Fig. 7a–c). Moreover, the GSH-to-GSSG ratio that was maintained close to control levels when worms fed with empty vector were grown in the presence of GA, glycine, or serine (Fig. 7d), was strongly reduced in animals defective in either *mel-32 or gcst-1*, independently of their supplementation with any of these three metabolites (Fig. 7d). In addition, animals defective in glycolate oxidation (*ghpr-1* and the *ghpr-1, ldh1, gox-1* mutants) exhibited a loss of response to glycine and L-serine when fed with *mel-32* RNAi bacteria (Supplementary Fig. 4). With this, we demonstrate that functional SHMT and GCS activities are essential for the observed GA-, glycine- and serine-mediated protection against oxidative damage. Supporting this conclusion, life-span extension caused by GA supplementation of worms treated with peroxide was lost in *mel-32*-knockdown animals (Supplementary Fig. 5).

Interesting to note, GA supplementation in the absence of H$_2$O$_2$ does not have a significant impact on any of the parameters analyzed (Figs. 2 and 3). This suggests that treatment with H$_2$O$_2$ could induce the expression of one or more of the proteins involved in the GA-metabolizing pathway (Fig. 4). In fact, it has long been established the role of H$_2$O$_2$ as inducer of the transcription factor Nrf-2 and its *C. elegans* ortholog Skn-1 that upregulate a plethora of phase II detoxification genes[47–51]. One of

the best known examples of genes induced by this transcription factor is *gcs-1* encoding glutamate cysteine ligase, the first enzyme in the biosynthesis of glutathione[31,50,52]. Therefore, we analyzed the relative expression levels of five different genes by RT-PCR upon peroxide or glycolate supplementation, including *gcs-1*. Remarkably, four of the tested genes (*gox-1*, *ghpr-1*, *mel-32*, and *gcs-1*) exhibited a significant induction when worms were exposed to different peroxide concentrations (Supplementary Fig. 6). The only exception was *gcst-1*, the expression of which was not significantly altered by the treatments (Supplementary Fig. 6). Unexpectedly, GA also moderately induced expression of these genes. With this, we demonstrate that both compounds contribute to the regulation of the GA-mediated antioxidant pathway. Although, their specific roles could differ.

**Glycolate supplementation restores the NADPH/NADP$^+$ ratio impaired by peroxide treatment.** Considering that both SHMT and GCS can produce 5,10-CH$_2$-THF, which is required for the regeneration of NADPH from NADP$^+$ (Fig. 4), the most plausible explanation for the increase in GSH/GSSG ratio caused by GA is the alteration of NADPH levels. NADPH is the major regulator of cellular redox potential and is crucial for the action of different antioxidant systems, including the maintenance of reduced glutathione pools[29,31,53] (Fig. 4). A series of recent publications has demonstrated that, in proliferating cells, the pentose phosphate pathway and the previously underestimated THF metabolism have nearly comparable contributions to the regeneration of NADPH from NADP$^+$[54–57] (Fig. 4). Thus, GA and consequently glycine could increase NADPH by entering serine–glycine metabolism.

When subjected to H$_2$O$_2$ treatment, *C. elegans* larvae showed a significantly decreased ratio of NADPH to NADP$^+$ (Fig. 8). However, when animals were supplemented with either GA or glycine in the presence of H$_2$O$_2$, the proportion of the reduced to the oxidized nucleotides was fully rescued (Fig. 8a), in agreement with the restoration of GSH/GSSG under the same conditions (Fig. 3c). L-serine supplementation, on the other hand, provoked a much weaker, but still significant, increase in the proportion of NADPH to NADP$^+$ (Fig. 8a). This observation cannot be explained by a reduced uptake of L-serine by the worms compared to that of GA or glycine since radioactivity measured in animals labeled with [$^{14}$C]L-serine was actually three times higher than in those labeled with [$^{14}$C]GA (Supplementary Fig. 7). Instead, the modest effect of L-serine on NADPH levels could be due to the fact that in *C. elegans*, SHMT is prone to synthesize serine from glycine[44]. Nevertheless, excessive amounts of exogenously added L-serine most probably cause the shift of equilibrium towards glycine and thus generate the improved reducing capacity of the worms. In addition, the conversion of L-serine to L-cysteine (Fig. 4) could also have a positive impact on GSH levels without affecting the relative amounts of NADP$^+$ and NADPH.

Consistent with the emerging contribution of one-carbon metabolism to the regeneration of NADPH, knocking down of the *C. elegans* SHMT enzyme, MEL-32, caused a reversal of the NADPH/NADP$^+$ ratio compared with worms fed with empty vector (Fig. 8b). Exposure of these *mel-32* knocked down worms to H$_2$O$_2$ produced a further decline in the proportion of the reduced to the oxidized nucleotides, which was not significantly affected by co-supplementation with GA. This correlates again with the ineffectiveness of GA addition on GSH/GSSG ratio in worms fed with *mel-32* RNAi (Figs. 7d and 8b). With this we conclude that glycolate exerts its beneficial effects on *C. elegans* exposed to oxidative stress via the glycine–serine pathway that

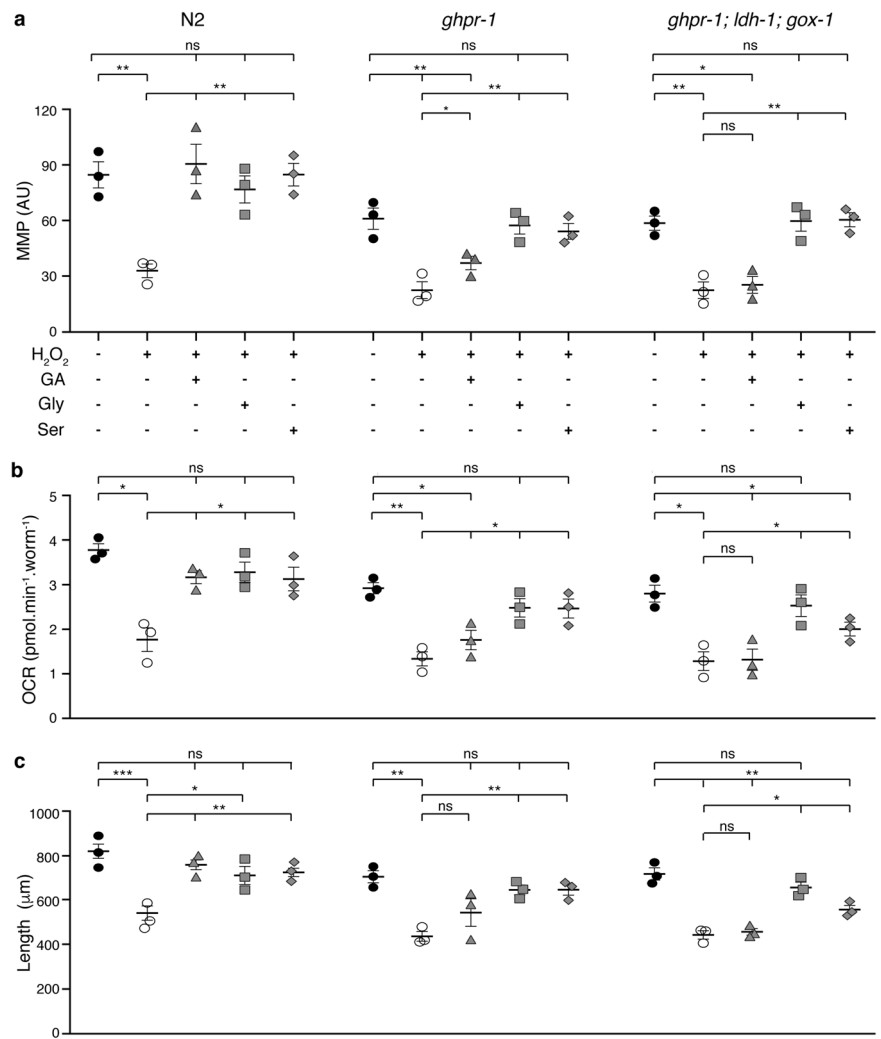

**Fig. 5 Glycine and L-serine supplementation rescue mitochondrial function and developmental rate of glycolate-oxidation-deficient worms.**
Mitochondrial membrane potential (**a**), respiration rates (**b**), and size (**c**) of wild type (N2) or mutant strains (*ghpr*−1 and *ghpr-1, gox-1, ldh-1*) exposed to 100 mM $H_2O_2$ and supplemented, with 10 mM glycolate, 100 μM glycine, or 500 μM L-serine, as indicated, as indicated. Measurements were carried out as described in Fig. 2. Error bars represent standard error of the mean ($n = 3$).

feeds one-carbon metabolism and the folate cycle, and in this way, improves the redox state of the animals.

## Discussion

Here, we have demonstrated that the endogenous metabolite glycolate works as an enhancer of the antioxidant response in *C. elegans* worms exposed to oxidative stress. In a previous report[19], we showed that the impairment of MMP caused by $PQ^{2+}$ treatment is reversed by glycolate. The present work, however, reveals that this compound cannot ameliorate all the traits affected by the pleiotropic herbicide. Therefore, we focused on a particular aspect of this treatment, that is the exposure to hydrogen peroxide, a major product of $PQ^{2+}$-induced oxidation (Fig. 9). We demonstrated that glycolate counteracts the deleterious effects of hydrogen peroxide at mitochondrial as well as cellular levels by entering serine–glycine metabolism. Furthermore, the observed recovery of the relative amounts of the reduced and oxidized forms of glutathione relies on glycolate-mediated regeneration of the redox status of the cells, thus improving the general state of the animal.

Our experiments show that the improvement in the antioxidant capacity of *C. elegans* caused by glycolate depends on two interlocked metabolic pathways: serine–glycine and one-carbon

metabolism. In recent years, hyperactivation of both have been proposed as possible drivers of oncogenesis[57–59]. Indeed, some cancer cells are highly dependent on serine/glycine uptake for proliferation, and these effects have been attributed to their incorporation as one-carbon units for purine synthesis and histone methylation. More recently, however, the fundamental role of these amino acids in the synthesis of NADPH has been brought to light[55–57]. For instance, knockdown of MTHFD (the enzyme converting 5,10-methylen-THF to THF while producing NADPH, Fig. 4) was shown to cause impaired resistance to peroxide with a decreased ratio of reduced to oxidized glutathione and a corresponding ROS accumulation[56]. Additionally, supporting our observations of increased levels of GSH and GSH/GSSG ratios upon glycolate addition (Fig. 3a, c), supplemental glycine has been reported to be rate-limiting to maintain glutathione levels in different animal studies[60–62] and even to promote longevity in *C. elegans*[44]. All in all, the current view of serine–glycine metabolism as a key player in the defense against oxidative stress is strengthened by the correlation we observed between functionality of this pathway and the protective power of glycolate in worms exposed to peroxide. Moreover, in agreement with the close interconnection between glycine and L-serine metabolism and their crucial role in the fine-tuning of

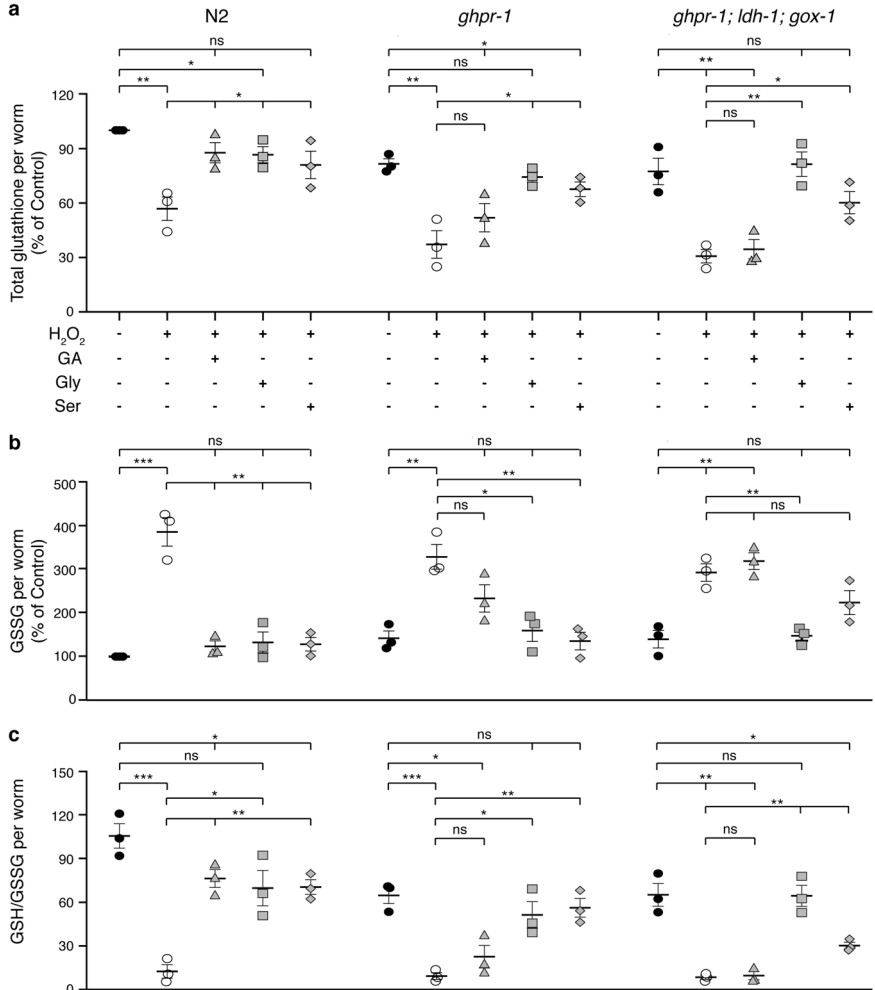

**Fig. 6 Glycine and L-serine but not glycolate can ameliorate the antioxidant capacity of *C. elegans* mutants unable to metabolize glycolate to glyoxylate. a** Total glutathione (GSH + GSSG) levels, **b** GSSG levels, and **c** ratio of GSH to GSSG in wild type (N2), single (*ghpr-1*), and triple (*ghpr-1, ldh-1, gox-1*) mutant animals exposed to 100 mM $H_2O_2$ and supplemented or not with 10 mM glycolate (GA), 100 µM glycine or 500 µM L-serine. Values were determined after 3 days of treatment and normalized to number of worms. Error bars represent standard error of the mean (n = 3).

central carbon pathways[57–59], supplementation with L-serine (a metabolite downstream of glycolate, Fig. 4) is not able to rescue *mel-32* or *gcst-1* defective animals (Fig. 7, Supplementary Fig. 4). Although these results are somewhat incongruent with the GA-metabolization pathway (Fig. 4), they can be explained by emerging view on the role of serine metabolism in the cell. Alterations in serine synthesis have been recently shown to disrupt mass balance within central carbon metabolism thus inducing metabolic disorders that are independent of serine utilization[63].

Key questions remain over where the enzymatic steps required for the antioxidant response occur, in the cytoplasm or in mitochondria, and whether both pools of glutathione can be tuned by glycolate supplementation. Of the three enzymes catalyzing the oxidation of glycolate to glyoxylate (Fig. 4), human GHPR-1 and LDH-1 are located in the cytoplasm, while the third one (GOX-1) is peroxisomal[35,39]. However, the next three reactions may occur mainly in the mitochondria (Fig. 4). Although in human cells the transamination of glyoxylate to glycine occurs in peroxisomes and mitochondria, in *C. elegans* it was reported to be limited to mitochondria[64]. GCS, has been reported to be associated with the inner mitochondrial membrane in animals and in plants[65]. Finally, there are two isoforms of SHMT in mammals, one cytoplasmic and the other mitochondrial[45,59,66], and the

same might be true for worms. However, given that the two isoforms of MTHFD are predicted to be cytoplasmic in *C. elegans*, the 5,10-$CH_2$-THF produced by GCS or SHMT (Fig. 4) should be shuttled to that compartment where the regeneration of NADPH could take place. Therefore, it is conceivable that glycolate supplementation would modify the redox balance mostly in the cytoplasm, thus altering the ratio of reduced to oxidized glutathione. It is possible that the mitochondrial glutathione pool might also be affected considering the existence of glutathione carriers through the mitochondrial membrane[67,68], thus explaining the improved mitochondrial activity we observed.

A remarkable observation described in our paper is that in worms not subjected to oxidative stress, glycolate addition does not alter the different parameters tested (Figs. 2 and 3). In this regard, a very interesting observation is provided by our RT-PCR experiments. Our finding that $H_2O_2$ can induce the expression of some genes encoding enzymes involved in the glycolate-mediated antioxidant pathway was foreseeable (*gox-1*, *ghpr-1*, *mel-32*, and *gcs-1*; Supplementary Fig. 6). Indeed, it has been extensively reported the peroxide-mediated activation of the transcriptional regulator Nfr-2 and its ortholog in *C. elegans*, Skn-1 that trigger the expression of antioxidant proteins and phase I and II detoxification enzymes[47,49–51,69]. Moreover, the Nrf-2-mediated upregulation of human *ghpr-1* and *mel-32* orthologs has been

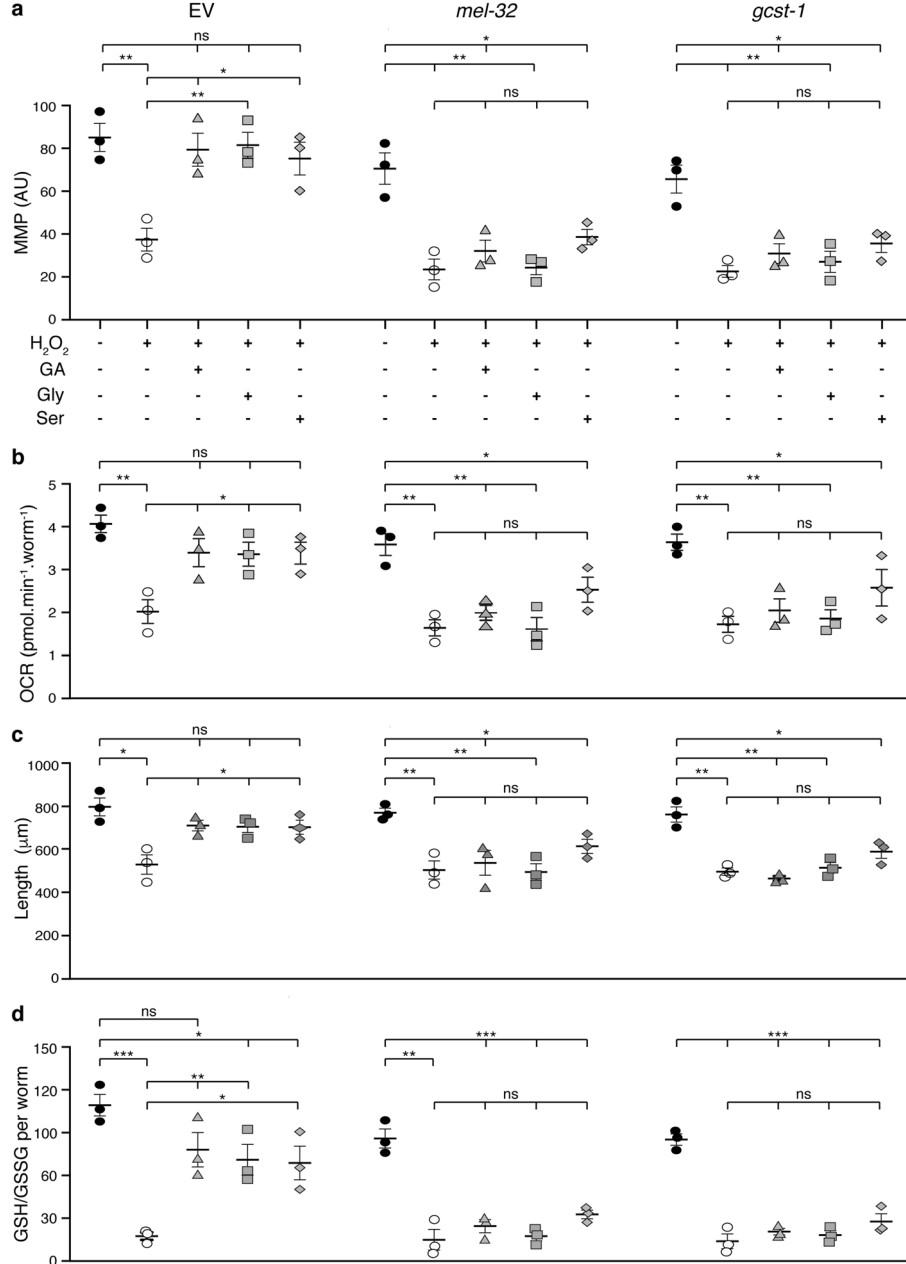

**Fig. 7 SHMT and GCS activities are essential for the restoration of mitochondrial function and antioxidant capacity mediated by glycine, L-serine, and glycolate upon peroxide treatment.** Mitochondrial membrane potential (**a**), respiration rates (**b**), size (**c**), and GSH/GSSG ratios (**d**) of wild type animals (N2) fed with empty vector (EV) or RNAi bacteria against *mel-32* or *gcst-1* genes exposed to 100 mM $H_2O_2$ and supplemented or not with 10 mM glycolate (GA), 100 µM glycine or 500 µM l-serine. Measurements were carried out as described in Figs. 2 and 3. Error bars represent standard error of the mean ($n = 3$).

recently reported in cancer cells as a mean to support glutathione and nucleotide production and to tune NADPH levels[70]. In the last years, several reports have shown that sublethal concentrations of ROS trigger a retrograde response that not only increases lifespan in different animal models, from worms to mammals, but also enhances "healthspan", particularly improving metabolism and immune system[51,71–73]. Surprisingly, however, our RT-PCR experiments revealed that GA induces expression of *gox-1, ghpr-1, mel-32,* and *gcs-1* to a similar extent as $H_2O_2$, although as was shown, glycolate alone had no effect on MMP, growth, or GSH levels. This counterintuitive result indicates that oxidative stress induced by $H_2O_2$ has more general impact on cellular processes than merely upregulation of the proteins involved in antioxidant

defense. In the absence of $H_2O_2$, the antioxidant action of glycolate is not required, although some enzymes for the metabolization of the latter can be elevated. Future studies should unravel molecular details of processes that enable $H_2O_2$ or its combination with glycolate to protect animals from oxidative damage.

In summary, our work provides a deeper understanding of the interplay between major metabolic pathways, mitochondrial activity, and organismal stress defense. Remarkably, the metabolic reactions described in our work are highly conserved throughout the eukaryotic kingdom. It is likely, therefore, that a similar outcome would be observed when supplementing glycolate, glycine, or serine in mammalian models of oxidative stress. This is particularly important because of the central role of these

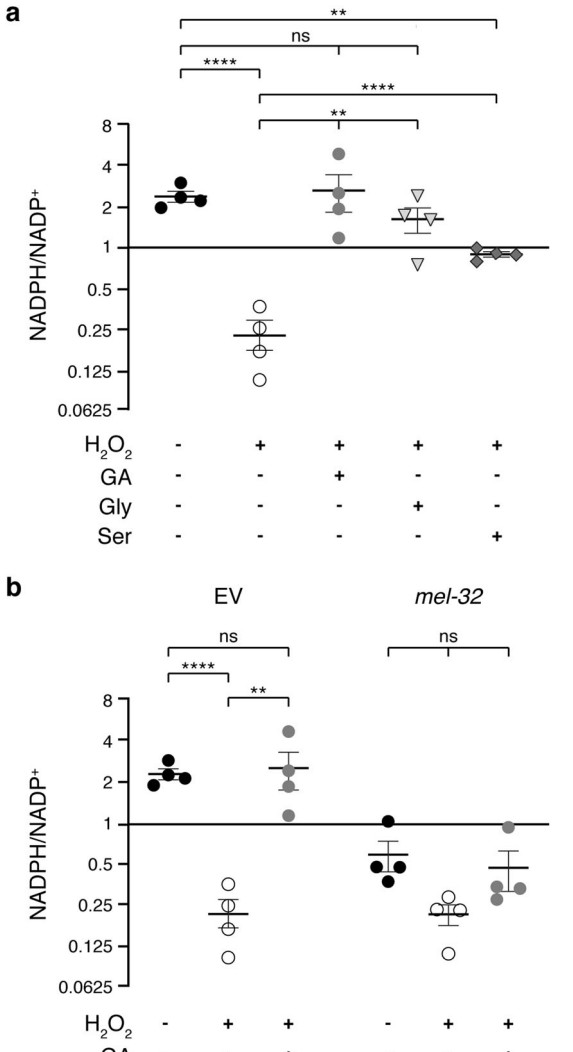

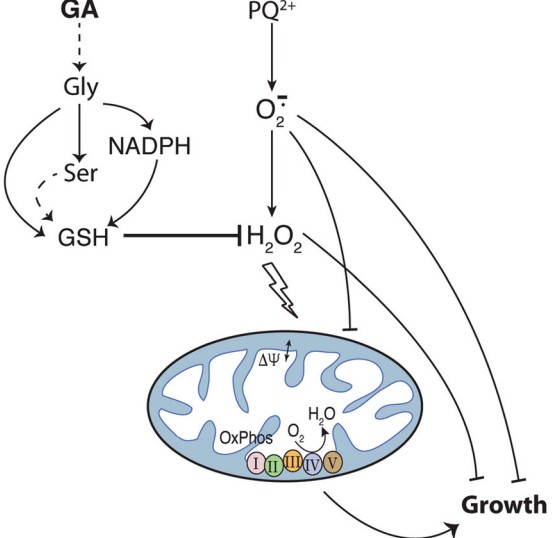

**Fig. 9 Scheme of the proposed pathway for the glycolate-mediated antioxidant activity.** Upon exposure to $PQ^{2+}$, superoxide ($O_2^{\cdot-}$) is produced that, in turn, can generate $H_2O_2$ (see text). Both ROS molecules disturb the mitochondrial activity (oxygen consumption rate and mitochondrial membrane potential, $\Delta\Psi$) and animal growth rate. Exogenously added glycolate (GA) is converted into glycine and serine. This triggers the $NADPH/NADP^+$ ratio that together with the generation of the building blocks for GSH synthesis, glycine and L-cysteine (the latter produced form L-serine), increase the GSH/GSSG ratio. The boosted GSH levels counteract the deleterious effects of $H_2O_2$ on mitochondrial function and growth but not those of superoxide itself.

**Fig. 8 Glycolate supplementation, via serine–glycine metabolism, increases the ratio of NADPH to NADP⁺ of peroxide-treated worms.** Ratio of NADPH to $NADP^+$ quantified by LC–MS and normalized to protein levels in samples in N2 worms treated as follows. **a** Animals were exposed to peroxide together with glycolate, glycine, or L-serine, as indicated, for 3 days. **b** Worms fed with empty vector (EV) or RNAi bacteria against SHMT (*mel-32*) genes were incubated with $H_2O_2$ alone or together with glycolate for 3 days. Error bars represent standard error of the mean ($n = 4$). 100 mM $H_2O_2$, 10 mM glycolate, 100 μM glycine, and 500 μM L-serine.

metabolic pathways in healthy development and defense, and becomes especially critical when considering the documented connection of many diseases with a decline in mitochondrial activity and oxidative stress[5,7,9,10,74]. Future studies may help to establish new therapeutic strategies based on glycolate, an endogenous antioxidant.

## Methods

**Chemicals.** Sodium glycolate (G0111, TCI), sodium D-lactate (71716, Sigma), glycine (G7126, Sigma), L-serine (S4500, Sigma), L-glutathione reduced (G4252, Sigma), L-buthionine sulfoximine (B2525, Sigma), hydrogen peroxide (H1009, Sigma), paraquat (sc-257968, SantaCruz biotechnologies or 36541 Fluka® from Sigma-Aldrich), IPTG (15529019, ThermoFisher) were used. [1–14C]-glycolate, [1–14C]-glycine, and L-[1–14C]-serine were purchased from HARTMANN ANALYTIC (Braunschweig, Germany).

**Worm strains and culture conditions.** C. *elegans* wild-type (N2) strain was received from *Caenorhabditis* Genetics Center, USA. Mutant strains were generated using the following primers. *ldh-1*: AATCAACAATTTTCATGTCT and TAAAAATCGCGCGCATTTGA; C31C9.2 (*ghpr-1*): TCTCGTATAAACAGAA AATATGG and GGGGCGCTCATTCTGGAAATTGG and F41E6.5 (*gox-1*): GAAGTTGCGTATGTCCTTCT and ATAATTGTTTCGAATCATGG. The injection protocol consisted of a master mix of 15 μl (5 μM of each of the primers, Cas9-Protein NLS [12.5 μM], tracrRNA-IDT [12.5 μM], dpy10 Oligo-IDT [733 nM], dpy-10-sgRNA-IDT [2,5 μM], and protein buffer-stock [2–3×]) that was injected into young adults of N2 strain. Progeny of the rescued strains were genotyped for homozygous deletion with PCR primers for three generations. All mutants were outcrossed at least twice with the wild type to eliminate background mutations. Double and triple mutants were obtained by crossing single or double mutants and selecting the homozygous progeny by PCR.

Worms were maintained at 20 °C on nematode growth medium (NGM) agar plates seeded with *Escherichia coli* NA22[75]. Gravid adults on NGM agar plates were treated with alkaline hypochlorite solution (i.e., bleached) to purify eggs. On the day prior to setting up the experiment, eggs obtained after bleaching were allowed to hatch overnight in M9 buffer to obtain synchronized L1 population.

For RNAi experiments in liquid, *E. coli* HT115 containing L4440 empty vector or the indicated cDNA fragment in L4440 was inoculated in 100 ml of LB containing 100 μg/ml ampicillin and incubated overnight at 37 °C in a shaking incubator at 250 rpm. To induce the production of dsRNA, a final concentration 0.8 mM IPTG was added to the collected bacterial pellet diluted 100-fold and incubated overnight at 25 °C in a shaking incubator at 250 rpm. Bacteria were then pelleted by centrifugation at 4000 rpm for 5 min and resuspended in 200 μl S-medium containing 100 μg/ml ampicillin and 0.8 mM IPTG.

**Paraquat treatment.** The treatment was performed as previously described[19]. Synchronized L1 larvae were added to NGM plates and treated with 200 μM $PQ^{2+}$ with or without 10 mM GA, 0.5 mM BSO, and 1 mM GSH for 2–7 days at 15 °C and compared to those grown in different conditions. Subsequently, they were prepared for mitochondrial staining, OCR, $H_2O_2$ quantification, or growth measurement.

**Peroxide treatment.** A population of synchronized L1 larvae was added to 30-ml glass tubes containing 2.5 ml of S-complete medium (liquid culture)[76] and NA22 or induced RNAi bacteria at an $OD_{600}$ of 1.9. 100 mM $H_2O_2$, 8 mM GA, 100 mM L-glycine, 500 mM L-serine, and 1 mM BSO were added as indicated. Worms were

incubated at 15 °C and 150 rpm in a shaking incubator for 2–6 days, refreshing the medium every 2 days.

**Mitochondrial staining and microscopy**. Mitochondrial staining was performed as described previously[19,77]. Briefly, L1 worms were grown on plate or in liquid culture for 2–3 days at 15 °C. Subsequently, they were pelleted in a 15-ml tube, rinsed twice with $H_2O$ and resuspended in 100 µl of M9. 5 µM MitoTracker Red CMXRos (Thermo Fisher Scientific, Germany) in DMSO was added to the worm suspension and incubated in this solution for 1.5 h at room temperature in the dark. Next, the excess dye was washed off with M9 buffer and worms were incubated in 100 µl M9 for 30 min to eliminate the excess dye in the gut. Finally, worms were paralyzed with 0.5 mM Levamizol (Sigma-Aldrich), placed on slides covered with a thin layer of 2% agarose on top of which the coverslip ($22 \times 22$ mm, Menzel-Glaser no. 1) was fixed using nail polish. We used a Zeiss LSM 700 inverted laser scanning confocal microscope and a Zeiss LCI Plan-Neofluar ×63/1.3 Imm Corr DIC M27 objective to image mitochondria (Zeiss, Germany). MitoTracker Red CMXRos was excited at 555 nm and the emission above 560 nm was acquired by the second PMT. Optical sections at $0.1 \times 0.1 \times 0.5$ µm$^3$ $x$–$y$–$z$ resolution were collected in a 4D hyperstack. Final images were adjusted for intensity and merged in Fiji. No non-linear adjustments were made.

**$H_2O_2$ measurements**. Hydrogen peroxide was measured using the Amplex Red hydrogen peroxide/peroxidase assay kit (Molecular Probes, Eugene, OR), adapting the protocol to *C. elegans*. In the presence of hydrogen peroxide, Amplex Red is oxidized by horseradish peroxidase to form a red-fluorescent oxidation product whose fluorescence intensity can be measured.

To measure ROS production from *C. elegans*, 5000–7000 L1 worms were exposed to each condition for 2–3 days as indicated and then washed four times in 1 ml of the reaction buffer supplied with the kit. Volumes were adjusted to 700–1200 nematodes/50 µl and 50 µl was pipetted into a 96-well plate. A total of 50 µl of the Amplex Red reaction buffer was then added to the wells, and after 1 h fluorescence was measured with a fluorescence microplate reader using excitation at $530 \pm 12.5$ nm and fluorescence detection at $590 \pm 17.5$ nm. Background fluorescence, determined for a no-$H_2O_2$ control reaction, was subtracted from each value. The significance of differences between conditions was determined by an unpaired $t$ test. GraphPadPrism 8.0 was used for these calculations. Three biological replicates were carried out for each condition.

**Growth rate experiments**. Synchronized L1 worms were incubated at 15 °C on plates or in liquid culture and exposed to different conditions as indicated. Single worms were selected randomly from the plates or aliquots were taken from the liquid cultures and added to NGM plates for immediate visualization. Twenty-five worms per treatment were measured at 24–48 h intervals and all growth curves were done four times. Images were analyzed using FIJI (Image J) software, with length measurements derived by the sum-total of a number of segmented lines of known length, down the center of the worm as described previously[78].

**Lifespan, progeny number, and embryonic lethality determination**. Lifespan experiments were performed at 20 °C without fluorouracil, as described elsewhere (Liu, 2019). Briefly, for the treatment during larval development worms were incubated in liquid culture under different conditions from L1 until L4, and then transferred onto control NGM plates until death. During the reproductive period (approximately days 1–8), worms were transferred to fresh plates every other day to separate them from their progeny. Survival was scored every other day throughout the lifespan and a worm was considered as dead when they did not respond to three taps. Worms that were missing, displaying internal egg hatching, losing vulva integrity, and burrow into NGM agar were censored. Progeny number and embryonic lethality was determined the day after transferring the adult animals to fresh plates. Statistical analyses of lifespan were calculated by log-rank (Mantel–Cox) tests on the Kaplan–Meier curves in GraphPad Prism. For progeny number and embryonic lethality, the significance of differences between conditions was determined using an unpaired $t$ test in GraphPadPrism 8.0. All experiments were performed in triplicate.

**Oxygen consumption assay**. Oxygen consumption of worms was measured with a Seahorse XF$^e$96 system (Seahorse Bioscience, North Billerica, MA), as previously described[77]. L3 larvae of the different strains were grown on plates or in liquid culture and supplemented for 2–4 days as described for each experiment. After removing bacteria and debris, ~100 worms were pipetted into each well of a 96-well Seahorse XF$^e$ assay plate and OCR was measured until it stabilized. Then, three subsequent measurements were made at 6.5-min intervals. Finally, the exact number of worms in each well was counted and used for normalization. A small number of abnormal readings were also filtered out at this stage. On average, 7–8 wells (technical replicates) were used for each condition. Normalized OCR values were averaged for the last three measurements for each strain and condition. Four biological replicates were analyzed for each condition.

**Quantification of glutathione**. Sample preparation was performed as described previously with some modifications[79]. From each condition ~10,000 synchronized

animals were harvested and washed three times with sterile water. Worms were resuspended in PBS buffer and 5-µl aliquots were taken to estimate the total number of worms, after which they were vortexed and frozen immediately with liquid nitrogen. To thoroughly lyse the worms, five cycles of freezing and thawing were performed in a water bath at 37 °C, followed by a 1-min sonication. The samples were then centrifuged and the supernatant tested for protein estimation and GSH quantification.

Total GSH, GSSG, and GSH/GSSG ratios were quantified using the GSH/GSSG-Glo assay kit (Promega) following the instructions of the manufacturer. Briefly, each sample was tested in triplicate in 96-well culture plates with total glutathione or GSSG lysis reagent, for total glutathione or GSSG measurement, respectively. The GSH probe, luciferin-NT, was converted to luciferin, which is coupled to a firefly luciferase. The plates were then read in an EnVision Luminometer plate reader (Perkin Elmer, United States). The GSSG value was subtracted from the total glutathione to calculate GSH levels and GSH/GSSG ratio. Four biological replicates were analyzed for each condition.

**NADPH and NADP$^+$ quantification**. To measure NADP$^+$ and NADPH from *C. elegans*, 10,000–15,000 L1 worms were exposed to each condition in duplicates for 3 days and then washed four times with $H_2O$. After taking an aliquot for protein estimation, worms were resuspended in 100 µl of $H_2O$ and 230 µl of Eluent A (95% acetonitrile, 0.1 mM ammonium acetate and 0.01% $NH_4OH$) was added together with 1 µl of 1 µM chloropropamide as internal standard. 25 µM solutions of each nucleotide were treated in parallel as standards. Homogenization for 15 min at 4 °C and $300 \times g$ was carried out in a TissueLyser II (Qiagen) after adding 1/3 volume of 0.5 mm zirconium beads. The resulting mixture was centrifuged at $13,000 \times g$ and the supernatant transferred to a new tube.

LC–MS/MS analysis of the samples was performed on a high-performance liquid chromatography (HPLC) system (*1200 Agilent*) coupled online to G2-S QTof (Waters). For normal phase chromatography Bridge Amide 3.5 ml ($2.1 \times 100$ mm) column from Waters was used. The mobile phase composed of eluent A and eluent B (40% acetonitrile, 0.1 mM ammonium acetate, and 0.01% $NH_4OH$) was applied with the following gradient program: Eluent B, from 0% to 100% within 18 min; 100% from 18 to 21 min; 0% from 21 to 26 min. The flow rate was set at 0.3 ml/min. The spray voltage was set at 3.0 kV and the source temperature was set at 120 °C. Nitrogen was used as both cone gas (50 L/h) and desolvation gas (800 L/h), and argon as the collision gas. MS$^E$ mode was used with Bridge Amide 3.5 µl ($2.1 \times 100$ mm) column in ESI-negative ionization polarity for the detection of the nucleosides. Mass chromatograms and mass spectral data were acquired and processed by MassLynx software (Waters). Three and four biological replicates were analyzed for each condition of *mel-32* RNAi and EV, respectively.

**Quantitative real-time PCR**. The differential expression of F41E6.5 (*gox-1*), C31C9.2 (*ghpr-1*), *gcst-1*, *mel-32*, and *gcs-1* genes was measured using qRT-PCR. ~5000 L1 N2 worms were exposed to each condition for 3 days at 15 °C as described above and then washed three times with $H_2O$. Total RNA was isolated using the RNeasy Mini Kit (Qiagen) and equal amounts of all samples were reverse transcribed with SuperScript III (Invitrogen) using oligo(dT)12–18 primers. cDNA quality was tested using standard PCR. qRT-PCR was performed using pre-designed TaqMan Gene Expression Assay (ThermoFischer Scientific) with at least three samples in triplicates. For the relative quantification of gene expression levels, the $2^{-\Delta\Delta CT}$ method and results were normalized to the expression level of the *cdc-42* gene. Expression levels were averaged over technical and biological replicates,

**Radioactive labeling**. To quantify the uptake of glycolate, glycine and L-serine ~10,000 synchronized L1 were incubated in liquid cultures as described above and grown for 3 days in the presence of 10 Ci of [1–$^{14}$C]-glycolate, [1–$^{14}$C]-glycine, or L-[1–$^{14}$C]-serine. At this stage, worms were snap-frozen and stored at 80 °C. Animals were lysed by freezing/thawing and extracted using the Bligh and Dyer method[80]. After phase separation, organic and aqueous fractions were recovered from the lower and upper phases, respectively. After drying and concentration, radioactivity was quantified using a scintillation counter and normalized for total number of worms.

**Statistics and reproducibility**. The number of biological replicates is stated in figure legends. Results shown as means; error bars represent the standard error of the mean. The unpaired Student's $t$-test was used to determine statistical significance of differences between means ($p < 0.05$ [*], $p < 0.01$ [**], $p < 0.005$ [***], $p < 0.0001$ [****]) unless otherwise stated.

**Reporting summary**. Further information on research design is available in the Nature Research Reporting Summary linked to this article.

## Data availability

All data supporting the main figures are available as Supplementary Data 1. All raw data files are being stored inhouse on backed-up servers and are available upon request to the corresponding author.

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

## Acknowledgements

We thank Sider Penkov (Institute for Clinical Chemistry and Laboratory Medicine, University Clinic and Medical Faculty, TU Dresden) for critical reading of the manuscript and constructive suggestions and other members of the Kurzchalia lab for fruitful discussions. We thank Iain Kennedy for editing the text and valuable advises.

## Author contributions

V.D. devised main conceptual ideas, conceived, designed and carried out experiments, collected and analyzed data, and wrote and edited the manuscript; S.T., K.S., and A.K.D.A. carried out experiments and analyzed data; T.V.K. devised main conceptual ideas, conceived experiments, edited, and made final approval of the manuscript.

## Funding

## Competing interests

The authors declare no competing interests.
