## [Peer Review File · Communications Biology]

Reviewers' comments:

Reviewer #1 (Remarks to the Author):

In this study, Diez et al. showed that glycolate, a compound that can restore the mitochondrial membrane potential of paraquat-treated worms, suppresses the deleterious effects of peroxide on mitochondrial activity and worm growth. They further showed that these actions of glycolate are through its entering into glycine/serine metabolism, which in turn enhances GSH/GSSG ratio and rescues H₂O₂ induced toxicity.

Overall this is an interesting study with solid data both in vivo and in vitro. The metabolic pathway that glycolate is involved is also adequately explored. However, additional studies are needed to improve this study.

Major points:

1. All the positive impacts of glycolate on worms were observed upon paraquat or H₂O₂ treatment. Have the authors analyzed the influences of glycolate alone on the life span, progeny, and embryonic lethality of N₂ worms (Figure 2D-2F)? If there are, whether these impacts are dependent on any of key enzymes they analyzed in the study (ghpr-1, idh-1, gox-1, mel-32, gcst-1)?
2. Along the same line, in Figure 3, it seems although GA rescued the GSH/GSSG ratio reduction upon H₂O₂ treatment, GA alone suppressed the GSH/GSSG ratio (Figure 3C). Have the authors checked whether ghpr-1, idh-1, gox-1, mel-32, or gcst-1 are induced by paraquat or H₂O₂ treatment?
3. In Figure S2, glycolate is not able to rescue paraquat induced defects on OCR and worm growth. Although it was mentioned that "This is consistent with the pleiotropic effects of this herbicide, which affects multiple targets, such as mitochondrial Complex I, aconitase, antioxidant defense, etc", can the authors provide further discussion on the possible toxicity mechanisms of paraquat that are independent of cellular redox homeostasis?
4. In Figure 6, the failure of serine to rescue the H₂O₂-induced mitochondrial toxicity as well as GSH/GSSG ratio in mel-32 or gcst-1 is surprising. In Figure 7A, serine supplementation was also only weakly increase the NADPH/NADP⁺. An explanation that the author offered is "This could be due to the fact that in *C. elegans*, SHMT is prone to synthesize serine from glycine" is not logically clear to me. Have the authors checked whether worms can efficiently uptake supplemented L-serine? Can the authors measure these amino acids in worms?
5. In the Discussion part, Page 14, the authors discussed about the "remarkable observation" that glycolate had on non-treated worms, and suggested that "the action of peroxide itself can induce or enhance the beneficial power of glycolate". I am not sure how to agree with this argument. To me this simply means that under normal condition there are not enough oxidative stress-induced mitochondrial toxicity that needs glycolate to defend. This also implies that glycolate is not harmful for worms under normal condition (need more characterization, as suggested in point #1), which may be a good sign for its therapeutic application. To prove their points that peroxide itself can induce the defense mechanism, the authors need to: (1) show whether the key enzymes that convert glycolate into NADPH/GSH are induced by H₂O₂, as suggested in point #2; (2) show that peroxide treated worms are better prepared to additional stress.

Reviewer #2 (Remarks to the Author):

The manuscript by Diez et al investigates the metabolic consequences of glycolate supplementation in *C. elegans*. This compound was previously known to have antioxidant capacity and protect

mitochondria from paraquat treatment. In the current manuscript the authors show that the glycolate mediated protection requires GSH. With solid supplementation and genetic experiments, they demonstrate that the glycine/serine/one carbon network is engaged by glycolate to affect the GSH/GSSG and NADPH/NADP⁺ ratios.

The conclusions of the paper are original and will be of interest to the wider field of redox biology. I personally appreciate the careful experimentation done to delineate the indirect mechanism of glycolate's antioxidant function.

Major points:

The authors show that the glycolate protective effect upon PQ treatment is specific to MMP toxicity, but that defects in OCR and growth caused by PQ are not corrected by glycolate. However, when peroxide is used as a stressor, all defects in all 3 phenotypes are suppressed. This needs clarification. Do the authors think that other PQ-induced stressors selectively affect OCR and growth, while peroxide, which is produced by PQ, has an overlapping effect?

For consistency across the paper, it would be great to repeat the experiments in Fig. 1 with peroxide. Also, what is the effect of a BSO control?

In the discussion the authors point out the surprising observation that GSH levels do not increase upon glycolate supplementation (Fig 3a). This is a major surprise in the presented data set and requires an explanation. The authors hypothesize that the oxidative stress is necessary to engage in glycolate utilization. It would be great to know, via qPCR or RNAseq, if indeed peroxide treatment is necessary for expression of the enzymes that utilize glycolate.

Minor points:

It appears that the survival curve in Fig 2 is a combination of the 3 lifespan replicates. I recommend to display only one independent survival experiment and to supply the data of all individual survival experiments in a supplemental table.

I think the data in Fig. S6 are very important for the paper and would recommend to move them to the main figure.

Glycolate is an endogenous metabolite – how does its change with age and how are its levels affected by external stress? In other words, are worms using this metabolite to buffer against potential stress? This goes beyond the paper but would be very exciting to understand.

Additionally, I wonder in which worm tissues glycolate is metabolized for the observed effects? Again, this goes beyond the scope but would be very interesting regarding cell-nonautonomous regulation of the antioxidant response.

Response to Reviewers' comments

COMMSBIO-20-1391-T

Glycolate combats massive oxidative stress by restoring redox potential in *Caenorhabditis elegans*

We thank all the reviewers for their constructive comments and feedback. We believe that performing suggested experiments and addressing their comments have strengthened and shaped our manuscript immensely. Below please find our point by point response in red.

Figures 1, 2, 3, 5, 6, 7 and 8 were modified following reviewers suggestions and Communications Biology formatting guidelines. All figures can be found at the end of this file.

Reviewer #1:

Major points:

1. All the positive impacts of glycolate on worms were observed upon paraquat or H₂O₂ treatment. Have the authors analyzed the influences of glycolate alone on the life span, progeny, and embryonic lethality of N2 worms (Figure 2D-2F)? If there are, whether these impacts are dependent on any of key enzymes they analyzed in the study (ghpr-1, idh-1, gox-1, mel-32, gcst-1)?

Thank you for the suggestion. In fact, we have analyzed life-span, progeny number and embryonic lethality on N2 worms treated only with glycolate, observing no statistically significant difference to those of untreated animals. This is now shown in Figure 2, panels D-F. Regarding the requirement of these key enzymes in the life-span extension observed by glycolate: We have quantified the lifespan of worms subjected to *mel-32* RNAi and treated with H₂O₂. Our results show that knock down of *mel-32* abolishes glycolate rescue, indicating that the extended survival upon peroxide treatment depends on this enzyme. We have added these data to Results section "Relief of oxidative stress requires entry of glycolate into serine-glycine metabolism" (lines 181-183) and in Supplementary Figure 5.

2. Along the same line, in Figure 3, it seems although GA rescued the GSH/GSSG ratio reduction upon H₂O₂ treatment, GA alone suppressed the GSH/GSSG ratio (Figure 3C). Have the authors checked whether ghpr-1, idh-1, gox-1, mel-32, or gcst-1 are induced by paraquat or H₂O₂ treatment?

Reduction of the GSH/GSSG ratio by GA alone (Figure 3) is not statistically significant. The beneficial effects of glycolate on peroxide-treated worms might be due to the H₂O₂-mediated induction of one or more enzymes involved in the postulated pathway. We have performed RT-PCR analysis of five enzymes and found that at least *gox-1*, *ghpr-1*, *mel-32* and *gcs-1* are elevated upon peroxide as well as glycolate treatment compared to untreated animals. These results are in agreement with the previously reported H₂O₂-mediated induction of the transcriptional regulator Nrf-2 and its *C. elegans* ortholog Skn-1 that upregulate a plethora of phase I and II detoxification genes^{3,5-7}. Moreover, supporting our results Denicola *et al.* (2015) have demonstrated the Nrf-2-mediated upregulation of PHGDH and SHMT2 (human orthologs of GHPR-1 and MEL-32, respectively) and tuning of NADPH levels in cancer cells⁸. This data is now shown in Supplementary Figure 6 and commented in results section "Relief of oxidative stress

requires entry of glycolate into serine-glycine metabolism” (lines 184-215) and in Discussion (lines 321-342).

3. In Figure S2, glycolate is not able to rescue paraquat induced defects on OCR and worm growth. Although it was mentioned that “This is consistent with the pleiotropic effects of this herbicide, which affects multiple targets, such as mitochondrial Complex I, aconitase, antioxidant defense, etc”, can the authors provide further discussion on the possible toxicity mechanisms of paraquat that are independent of cellular redox homeostasis?

Thank you for the comment. Indeed, several mechanisms have been proposed for the H₂O₂-independent toxic effects of Paraquat. Among them is the NADPH consumption by the redox cycling of Paraquat, what could prevent several anabolic processes (i.e. biosynthesis of fatty acids, lipoprotein and amino acids). In addition, superoxide (that is upstream of H₂O₂ on the paraquat cascade) can directly affect Fe-S cluster-containing enzymes, thus releasing excessive amount of Fe²⁺ to the cells causing cytotoxicity. This has now been added to Results section (lines 24-28 and 31-33).

4. In Figure 6, the failure of serine to rescue the H₂O₂-induced mitochondrial toxicity as well as GSH/GSSG ratio in *mel-32* or *gcst-1* is surprising. In Figure 7A, serine supplementation was also only weakly increase the NADPH/NADP⁺. An explanation that the author offered is “This could be due to the fact that in *C. elegans*, SHMT is prone to synthesize serine from glycine” is not logically clear to me. Have the authors checked whether worms can efficiently uptake supplemented L-serine? Can the authors measure these amino acids in worms?

Indeed, we did not have any prove about the relative uptake of glycolate, glycine and L-serine by *C. elegans* in our experimental conditions. Therefore, we have now performed labelling of *C. elegans* with radioactive glycolate, glycine or L-serine and incubated the worms in the presence or absence of H₂O₂. Our experiments demonstrate that L-serine and glycine are taken up with higher efficiency than glycolate (Supplementary Figure 7) and lines 233-236. These observations thus cannot explain the lack of positive effect of L-serine on *gcst-1* and *mel-32* mutants as well as on NADPH/NADP⁺ ratio. Instead, we postulate that serine synthesis alterations (like those caused by *gcst-1* and *mel-32*) could disrupt mass balance within central carbon metabolism thus inducing metabolic disorders that are independent of serine utilization. This was already suggested by Reid *et al.* (2018)⁹. Moreover, Labuschagne *et al.* (2014) showed that although serine and glycine can be interconverted, exogenous glycine cannot replace serine to support cancer cell proliferation¹⁰. The close interconnection between glycine and L-serine metabolism and their crucial role in the fine-tuning of the central carbon pathways^{11,12} is now more in details explained in the lines 237-246 and 290-297).

5a. In the Discussion part, Page 14, the authors discussed about the “remarkable observation” that glycolate had on non-treated worms and suggested that “the action of peroxide itself can induce or enhance the beneficial power of glycolate”. I am not sure how to agree with this argument. To me this simply means that under normal condition there are not enough oxidative stress-induced mitochondrial toxicity that needs glycolate to defend.

Thank you for your suggestion. Indeed, this hypothesis could explain our RT-PCR results, were not only H₂O₂ but also glycolate are inducing glycolate-metabolization enzymes (Supplementary Figure 6). This implies that even when these key enzymes are upregulated by glycolate, oxidative stress in the absence of H₂O₂ is below certain threshold and therefore there is not target to be improved. We have now added this possibility to Discussion (lines 237-240).

5b. This also implies that glycolate is not harmful for worms under normal condition (need more characterization, as suggested in point #1), which may be a good sign for its therapeutic application. To prove their points that peroxide itself can induce the defense mechanism, the authors need to: (1) show whether the key enzymes that convert glycolate into NADPH/GSH are induced by H₂O₂, as suggested in point #2; This issue is addressed in point #2.

5c. (2) show that peroxide treated worms are better prepared to additional stress. Several reports have shown the existence of endogenous metabolites that promote cellular and organismal resilience through the induction of an adaptive mechanisms commonly termed mitohormesis. These endogenous compounds (among them, sublethal concentrations of ROS) trigger a retrograde response that not only increase lifespan in different animal models, from worms to mammals but also enhance “healthspan”, particularly improving metabolism and immune system¹⁻⁴. This is now commented in Discussion (lines 326-334).

Reviewer #2 (Remarks to the Author):

The manuscript by Diez et al investigates the metabolic consequences of glycolate supplementation in *C. elegans*. This compound was previously known to have antioxidant capacity and protect mitochondria from paraquat treatment. In the current manuscript the authors show that the glycolate mediated protection requires GSH. With solid supplementation and genetic experiments, they demonstrate that the glycine/serine/one carbon network is engaged by glycolate to affect the GSH/GSSG and NADPH/NADP⁺ ratios.

The conclusions of the paper are original and will be of interest to the wider field of redox biology. I personally appreciate the careful experimentation done to delineate the indirect mechanism of glycolate’s antioxidant function.

Major points:

1. The authors show that the glycolate protective effect upon PQ treatment is specific to MMP toxicity, but that defects in OCR and growth caused by PQ are not corrected by glycolate. However, when peroxide is used as a stressor, all defects in all 3 phenotypes are suppressed. This needs clarification. Do the authors think that other PQ-induced stressors selectively affect OCR and growth, while peroxide, which is produced by PQ, has an overlapping effect? The same point was raised by Reviewer #1. Indeed, several mechanisms have been proposed for the H₂O₂-independent toxic effects of Paraquat. Among them is the NADPH consumption by the redox cycling of Paraquat, what could prevent several anabolic processes (i.e. biosynthesis of fatty acids, lipoprotein and amino acids). In addition, superoxide (that is upstream of H₂O₂ on the paraquat cascade) can directly affect Fe-S cluster-containing enzymes, thus releasing excessive amount of Fe²⁺ to the cells causing cytotoxicity. This has now been added to Results (lines 24-28 and 31-33).

2. For consistency across the paper, it would be great to repeat the experiments in Fig. 1 with peroxide. In Figure 2 we present experiments performed with peroxide, including BSO treatment similarly to Figure 1. Moreover, in this Figure we included not only MMP but also other parameters that are rescued by GA treatment (OCR, growth, lifespan, progeny number and H₂O₂ levels) and the BSO effect in each case. Unexpectedly, GSH supplementation of worms exposed to H₂O₂ is deleterious for the worms and therefore, it was not possible to test the different parameters in the presence of both BSO and GSH upon H₂O₂ treatment (lines 67-68). Presently, we do not have any reasonable explanation for this phenomenon but it would be very interesting to understand this in the future. 2b. Also, what is the effect of a BSO control? Upon BSO treatment levels of glutathione were decreased but restored when GSH was included in the medium observing, at the same time, a correlation between the tripeptide amounts and MMP. This effect has now been added to Supplementary Figure 1 and described in the result section “Glycolate restores paraquat-mediated reduction in mitochondrial membrane potential via a GSH-dependent mechanism” (lines 15-18).

3. In the discussion the authors point out the surprising observation that GSH levels do not increase upon glycolate supplementation (Fig 3a). This is a major surprise in the

presented data set and requires an explanation. The authors hypothesize that the oxidative stress is necessary to engage in glycolate utilization. It would be great to know, via qPCR or RNAseq, if indeed peroxide treatment is necessary for expression of the enzymes that utilize glycolate.

Thank you for the suggestions. We have now performed RT-PCR of 5 genes involved in the postulated pathway observing that *gox-1*, *ghpr-1*, *mel-32* and *gcs-1* are significantly upregulated by H₂O₂ treatment as well as by glycolate. From this, we postulate that under control condition oxidative stress-induced mitochondrial toxicity is not high enough to require glycolate protection, as also suggested by Reviewer#1. However, upon H₂O₂ treatment toxicity produced by oxidative stress can be ameliorated by glycolate supplementation. These observations are now commented in lines 184-215 and 321-342) and shown in Supplementary Figure 6.

Minor points:

4. It appears that the survival curve in Fig 2 is a combination of the 3 lifespan replicates. I recommend to display only one independent survival experiment and to supply the data of all individual survival experiments in a supplemental table.

Indeed, we have noticed that some of the data from one of the curves were missing. We have combined all lifespan replicates with the corresponding error bars because we consider it is a more accurate representation of how the system works (Figure 2).

5. I think the data in Fig. S6 are very important for the paper and would recommend to move them to the main figure.

We have moved this Supplementary Figure to main text as Figure 6.

6. Glycolate is an endogenous metabolite – how does it change with age and how are its levels affected by external stress? In other words, are worms using this metabolite to buffer against potential stress? This goes beyond the paper but would be very exciting to understand.

According to our data not shown, glycolate itself is a transient metabolite, most probably converted immediately to glycine. We hypothesize that the latter is buffered, instead. This is indeed a very exciting topic and some of the questions have been addressed recently^{13,14} where glycine levels have been quantified along larval stages and adulthood and correlated them to worm longevity.

7. Additionally, I wonder in which worm tissues glycolate is metabolized for the observed effects? Again, this goes beyond the scope but would be very interesting regarding cell-nonautonomous regulation of the antioxidant response.

Study of single cell metabolism in *C. elegans* has not been established yet but it is a very interesting topic indeed. This is an exciting issue to be analyzed in the future.

1. Miranda-Vizueté, A. & Veal, E. A. *Caenorhabditis elegans* as a model for understanding ROS function in physiology and disease. *Redox Biol.* **11**, 708–714 (2017).
2. Fischer, F. & Ristow, M. Endogenous metabolites promote stress resistance through induction of mitohormesis. *EMBO Rep.* **21**, (2020).
3. Ristow, M. & Schmeisser, K. Mitohormesis: Promoting health and lifespan by increased levels of reactive oxygen species (ROS). *Dose-Response* **12**, 288–341 (2014).
4. Matsumura, T. *et al.* N-acetyl-L-tyrosine is an intrinsic triggering factor of mitohormesis in stressed animals. *EMBO Rep.* **21**, (2020).
5. Jiang, Z. Y., Lu, M. C. & You, Q. D. Nuclear Factor Erythroid 2-Related Factor 2 (Nrf2) Inhibition: An Emerging Strategy in Cancer Therapy. *J. Med. Chem.* **62**, 3840–3856 (2019).
6. Blackwell, T. K., Steinbaugh, M. J., Hourihan, J. M., Ewald, C. Y. & Isik, M. SKN-1/Nrf, stress responses, and aging in *Caenorhabditis elegans*. *Free Radic. Biol. Med.* **88**, 290–

- 301 (2015).
7. Naji, A. *et al.* The activation of the oxidative stress response transcription factor SKN-1 in *Caenorhabditis elegans* by mitis group streptococci. *PLoS One* **13**, 1–19 (2018).
 8. Denicola, G. M. *et al.* NRF2 regulates serine biosynthesis in non-small cell lung cancer. *Nat Genet* **47**, 1475–1481 (2015).
 9. Reid, M. A. *et al.* Serine synthesis through PHGDH coordinates nucleotide levels by maintaining central carbon metabolism. *Nat. Commun.* **9**, 1–11 (2018).
 10. Labuschagne, C. F., van den Broek, N. J. F., Mackay, G. M., Vousden, K. H. & Maddocks, O. D. K. Serine, but not glycine, supports one-carbon metabolism and proliferation of cancer cells. *Cell Rep.* **7**, 1248–1258 (2014).
 11. Yang, M. & Vousden, K. H. Serine and one-carbon metabolism in cancer. *Nat. Rev. Cancer* **16**, 650 (2016).
 12. Locasale, J. W. Serine, Glycine and the one-carbon cycle: cancer metabolism in full circle. *Nat. Rev Cancer* **13**, 572–583 (2013).
 13. Gao, A. W. *et al.* A sensitive mass spectrometry platform identifies metabolic changes of life history traits in *C. elegans*. *Sci. Rep.* **7**, 1–14 (2017).
 14. Liu, Y. J. *et al.* Glycine promotes longevity in *caenorhabditis elegans* in a methionine cycle-dependent fashion. *PLoS Genet.* **15**, 1–23 (2019).

Figures

Figure 1. A

Figure 2.

Figure 3.

Figure 4.

Figure 5.

Figure 6.

Figure 7.

Figure 8.

Figure 9.